# Proline-Induced Modifications in Morpho-Physiological, Biochemical and Yield Attributes of Pea (*Pisum sativum* L.) Cultivars under Salt Stress

Sadia Shahid [1], Muhammad Shahbaz [1], Muhammad Faisal Maqsood [2], Fozia Farhat [3,*], Usman Zulfiqar [4], Talha Javed [5,6,*], Muhammad Fraz Ali [5], Majid Alhomrani [7,8] and Abdulhakeem S. Alamri [7,8]

1 Department of Botany, University of Agriculture, Faisalabad 38000, Pakistan
2 Department of Botany, The Islamia University of Bahawalpur, Bahawalpur 63100, Pakistan
3 Department of Botany, Government College Women University, Faisalabad 38000, Pakistan
4 Department of Agronomy, Faculty of Agriculture and Environment, The Islamia University of Bahawalpur, Bahawalpur 63100, Pakistan
5 Department of Agronomy, University of Agriculture, Faisalabad 38040, Pakistan
6 College of Agriculture, Fujian Agriculture and Forestry University, Fuzhou 350002, China
7 Department of Clinical Laboratories Sciences, The Faculty of Applied Medical Sciences, Taif University, Taif 21944, Saudi Arabia
8 Centre of Biomedical Sciences Research (CBSR), Deanship of Scientific Research, Taif University, Taif 21944, Saudi Arabia
* Correspondence: foziafarhat@gcwuf.edu.pk (F.F.); talhajaved54321@gmail.com (T.J.)

**Abstract:** Climate change is aggravating soil salinity, causing huge crop losses around the globe. Multiple physiological and biochemical pathways determine the ability of plants to tolerate salt stress. A pot experiment was performed to understand the impact of proline levels, i.e., 0, 10, 20 mM on growth, biochemical and yield attributes of two pea (*Pisum sativum* L.) cultivars (cv. L-888 and cv. Round) under salt stress (150 mM) along with control (0 mM; no stress). The pots were filled with river-washed sand; all the plants were irrigated with full-strength Hoagland's nutrient solution and grown for two weeks before application of salt stress. Foliar spray of proline was applied to 46-day-old pea plants, once a week till harvest. Data for various growth and physio-biochemical attributes were collected from 70-day-old pea plants. Imposition of salt stress significantly checked growth, gas exchange characteristics [net $CO_2$ assimilation rate ($A$), transpiration rate ($E$), stomatal conductance ($g_s$)], total soluble proteins, concentration of superoxide dismutase (SOD), shoot and root $K^+$ and $Ca^{2+}$ contents, while sub-stomatal $CO_2$ concentration ($Ci$), coefficient of non-photochemical quenching ($q_N$), non-photochemical quenching (NPQ), concentration of catalase (CAT) and peroxidase (POD), free proline, and shoot and root $Na^+$ contents increased substantially. Foliar application of proline significantly improved growth, yield, $A$, $g_s$, activity of POD, and shoot and root $K^+$ and $Ca^{2+}$ contents, while decreased NPQ values in both pea cultivars under stress and non-stress conditions. Moreover, both pea cultivars showed significant differences as cv. Round exhibited a higher rate of growth, yield, gas exchange, soluble proteins, CAT activity, free proline, shoot and root $K^+$ and $Ca^{2+}$ contents compared to L-888. Hence, the outcomes of this study pave the way toward the usage of proline at 20 mM, and cv. Round may be recommended for saline soil cultivation.

**Keywords:** antioxidants; gas exchange; proline; salinity; stress endorsement

## 1. Introduction

Salinity or salinization (especially secondary salinization) is not a recent global phenomenon resulting from expanding urbanization, industrialization or the modernization of agriculture, but an age-old problem of irrigated agriculture [1]. However, salinity is becoming more severe with the expanding population and displays serious threats to land under cultivation around the globe, reducing the capacity of all forms of the terrestrial

ecosystems by lowering our biodiversity, agricultural productivity, damaging the environment, contaminating groundwater, creating flood risks and food security issues, and limiting the economic growth of a community [2]. About 33% of cultivated lands are categorized as salt-affected soil, which may increase up to 50% by 2050 [3]. This trend grows antagonistic with the ever-increasing challenge of ensuring global food security, and so creates an emergent situation in which there is a need to search for more cultivated land and enhance crop productivity even in barren soil by establishing efficient and tolerant crops capable of growing in salty conditions [4,5]. The global agriculture sector currently encounters many challenges, including the production of 70% more food to ensure food security [6]. In many situations, lower productivity is caused by various environmental stresses [7]. The more intricately regulated stresses such as high salinity, drought, cold, and heat have a negative effect on the survival, development, biomass and yield of economically important food crops [8].

The over-accumulation of salinity in the plant's rhizosphere, which imposes highly deleterious effects on plant biomass [9], physiology [10], accumulation of mineral ions [11,12], damage PSII reaction centers [13], and metabolic dysfunction because of production of reactive oxygen species (ROS), leads to growth retardation with substantial loss of metabolic functioning of the plant [14]. Notably, many plants attempt to balance both enzymatic and non-enzymatic antioxidant defense systems under extreme salinity stresses [15].

Pea (*Pisum sativum* L.) is a leguminous crop that is cultivated in tropical and subtropical areas of the world [16]. It is a cool season vegetable crop that is used as food and fodder throughout the country. The pea is an excellent source of various vitamins, minerals, antioxidants, salts, carbohydrates and proteins [17]. The field pea is a highly significant pulse crop. Canada was ranked first in area (21%) and production (35%) at global level, while China occupies the second position in area (13.70%), followed by Russia (12.94%). It is worth mentioning that in spite of low area utilized for pea cultivation, Ireland has the highest productivity (5000 kg ha$^{-1}$), followed by the Netherlands (4766 kg ha$^{-1}$), and Denmark (4048 kg ha$^{-1}$). In developed countries, the field pea is grown on industrial scale, whereas in developing countries, these are grown on subsistence level and considered a staple food [18].

The pea is known to possess moderate tolerance to salinity stress [19]. Among various environmental constraints, soil salinity adversely affects growth, gas exchange, activities of antioxidant enzymes and mineral nutrients of pea plants [20]. The high concentration of antioxidants under salt stress conditions can cause damage and correlate it significantly with plant tolerance against stress [21]. Similarly, salinity could also result in the display of many other abiotic stresses and different physiological abnormalities in plants [22]. Exogenous application of low molecular weight compounds such as glycine betaine and proline can reduce damaging effects of salt stress by enhancing plant's tolerance against abiotic stresses [23,24]. These molecules rescue the plants and regulate osmotic adjustment to enhance salt stress tolerance in plants [25]. Proline acts as an osmolyte, a metal chelator and a signaling molecule. Proline maintains the structure of membranes, prevents electrolyte leakage and reduces the level of reactive oxygen species [26,27]. Proline reduces membrane damages by decreasing oxidative stress attributes such as MDA contents in cucumber [28] and $H_2O_2$ [29]. These molecules help the plants to regulate osmotic adjustment and to enhance abiotic stress tolerance in plants. Exogenous application of proline enhances stress tolerance in different food crops when applied in an appropriate concentration e.g., cucumber (*Cucumis sativus* L.) [30], rice (*Oryza sativa* L.) [31,32], wheat (*Triticum aestivum* L.) [33], and sunflower (*Helianthus annuus* L.) [34].

Proline has a diversified role in plants, particularly under stressful conditions. However, there is no or very little information available about foliar application of proline on pea plants under salt stress. Regarding the significance of proline, it is hypothesized that foliar application of proline could mitigate deleterious effects of salt stress and enhance growth, gas exchange, chlorophyll fluorescence, antioxidant defense systems, mineral nutrients and yield potential of pea plants under saline or non-saline conditions. Furthermore, the

different levels of proline were critically analyzed on pea plants with and without salt stress proline.

## 2. Materials and Methods

In order to investigate the effect of foliar application of proline on two pea cultivars (L-888 and Round), a pot experiment was conducted under natural climatic conditions at the University of Agriculture, Faisalabad, Punjab, Pakistan. Seeds of two cultivars were obtained from Ayub Agricultural Research Institute (AARI), Faisalabad, Punjab, Pakistan. Round pots with specific dimensions (top width, 27 cm; bottom width, 20.5 cm) were used for the current project. There were six sets of 36 pots: one set for control, the second set for 10 mM Pro application, the third set for 20 mM Pro application, the fourth set for salt stress, the fifth set of salt + 10 mM Pro, and the sixth set for salt + 20 mM Pro. Eight seeds were sown in each plastic pot in thoroughly washed river sand (5.5 kg per pot). After germination, 5 seedlings per pot were maintained. Plants were nourished with full-strength Hoagland's nutrient solution to attain the plant's essential nutritional requirements until the application of salt stress (i.e., 14 days). After two weeks of germination, two regimes of salt stress were maintained as non-saline. Only full-strength Hoagland's nutrient solution and saline (150 mM NaCl in full strength Hoagland's nutrient solution) were applied and continued until the harvest. Salinity level (150 mM) in the rhizosphere was achieved by gradually increasing the salt concentration in aliquots of 50 mM NaCl for three days to avoid osmotic shock. Proline (Mol. wt. 115.11; Sigma Aldrich, Waltham, MA, USA), was used for making the different proline concentrations (0, 10 and 20 mM + 0.1% tween-20). Proline solutions were prepared according to standard protocol. A stock solution of proline (30 mM) was prepared in deionized water, later diluted to desired concentration for the experiment. Tween-20 was used as surfactant. Foliar application of three proline levels was performed at 25 mL per pot to 56-day-old pea plants. Design of experiment was completely randomized with four replicates. Fresh leaf samples of 70-day-old pea plants were collected in plastic zipper bags, stored at $-20\,^{\circ}$C and used for the determination of various growth and physicochemical attributes.

### 2.1. Determination of Growth Attributes

Two plants from each pot were carefully uprooted; their shoot/root lengths and fresh weights were measured. The plants were oven-dried at 65 $^{\circ}$C up to their constant dry weight.

### 2.2. Gas Exchange Characteristics

Infrared gas analyzer (IRGA) of Analytical Development Company, LCA-4 ADC, Hoddesdon, was used for the determination of gas-exchange characteristics such as net $CO_2$ assimilation rate ($A$), transpiration rate ($E$), stomatal conductance ($g_s$) and sub-stomatal $CO_2$ ($C_i$). Data were recorded according to the specifications as described in [35] from 11:00 a.m. to 01:00 p.m. with Qleaf (PAR) value as 942 µmol m$^{-2}$s$^{-1}$.

### 2.3. Determination of Chlorophyll Fluorescence Attributes

For the determination of chlorophyll fluorescence attributes, a multi-mode chlorophyll fluorometer (OS5P-Sciences, Inc. Winn Avenue, Hudson, NH, USA) was used according to the method of [36] with specifications as described by Nusrat [20].

### 2.4. Determination of Soluble Proteins

Total soluble protein contents in fresh leaf samples were determined by the [37]. Leaf samples (0.5 g) were finely homogenized in phosphate buffer (50 mM; pH 7.8) in an ice bath. Samples were then centrifuged 12,000× $g$ for 10 min at 4 $^{\circ}$C, collected 1 mL supernatant in a clean glass test tube and mixed with 5 mL Bradford reagent. Samples were incubated for 15 min at room temperature and determined protein contents with a spectrophotometer (IRMECO, 2020, Lütjensee, Germany) at 595 nm.

### 2.5. Determination of Antioxidant Enzymes Activities

Enzyme extract was collected by homogenizing fresh leaf samples in 10 mL of phosphate buffer (50 mM; pH 7.8) and centrifuged at $15,000 \times g$ for 20 min at 4 °C.

Giannopolitis and Ries [38] method was used to appraise the activity of SOD via inhibition in photoreduction of nitrobluetetrazolium (NBT). The amount of enzyme that inhibits 50% of NBT is considered equivalent to one unit of SOD activity. Reaction mixture consists of 400 µL of distilled $H_2O$ + 250 µL buffer (pH 7.8) + 100 µL methionine + 50 µL NBT + 50 µL leaf extract and 50 µL riboflavin. Plastic cuvettes containing reaction mixture were kept under light for 20 min. and read OD at 560 nm with a spectrophotometer.

For the determination of CAT and POD activities, the Chance and Maehly [39] method was used. The reaction mixture was prepared by adding 50 mM phosphate buffer (1.9 mL) + 5.9 mM $H_2O_2$ (1 mL) in 100 µL enzyme extract. Change in CAT activity was determined after every 20 s for 2 min. at 240 nm with spectrophotometer. For POD determination reaction solution contain 750 µL phosphate buffer (50 mM), 100 µL guaiacol (20 mM), 100 µL $H_2O_2$ (40 mM) and 100 µL enzyme extract. For 3 min. absorbance of reaction mixture was read after every 20 s at 470 nm with spectrophotometer.

### 2.6. Determination of Free Proline Contents

Fresh leaf samples (0.5 g) were finely homogenized in 10 mL of 3% sulfosalicylic acid (0.14 M), centrifuged $15,000 \times g$ for 20 min. The 2 mL supernatant was mixed with 2 mL acid ninhydrin (pH 3.6) and 2 mL glacial acetic acid (pH 2.4) and incubated at 95 °C for 60 min. Next cooled the samples immediately, added 4 mL of toluene, vortexed and read absorbance of chromophore layer 520 nm with spectrophotometer according to the method of Bates [40].

### 2.7. Mineral Ions Determination

Shoot and root mineral contents were determined according to the method of Allen [41]. To 0.1 g finely grinded shoot or root material were separately digested in concentrated sulfuric acid in 50 mL digestion flasks and incubated overnight at room temperature. Samples were then placed on a hot plate, the temperature gradually increased to 250 °C until fumes started liberating. Then 0.5 mL of $H_2O_2$ (35%) was added in the containing flasks until samples became colorless. Cooled the samples, increased volume up to 50 mL with distilled $H_2O$ and filtered. Filtrate was then used for the determination of various ions such as $Na^+$, $K^+$ and $Ca^{2+}$ with the help of a flame photometer from the Jenway (PEP 7) company (Vernon Hills, IL, USA).

### 2.8. Statistical Analysis of Data

Data of various growth and physiochemical parameters were analyzed using Co-STAT computer program for analysis of variance (ANOVA) determination. For comparing mean values least significant difference was used according to the method of Snedecor and Cochran [42].

## 3. Results

### 3.1. Effect of Salinity Stress and Proline on Growth and Yield Attributes of Pea Seedlings

Shoot fresh/dry weight, shoot length and root length were reduced significantly ($p \leq 0.001$) under salinity stress compared to control plants (Table 1). On the other hand, all studied morphological features tended to enhance with foliar application of Pro (Figure 1). Both pea cultivars showed non-significant difference ($p \leq 0.05$) with respect to saline and proline treatment, except for shoot dry weight (Figure 1b) and shoot length (Figure 1e, Table 1). Salinity stress decreased the shoot fresh and dry weight by 13.5% and 37.7%, and root fresh and dry by 5.72% and 10.7%, respectively, compared to the control (Figure 1a–d). Similarly, shoot length and root length showed 12% and 2.3% decline under salt stress, respectively (Figure 1e,f). The foliar application of Pro displayed stimulatory effects on growth parameters to mitigate the saline toxicity. The foliar spray of 20 mM Pro increased

the shoot fresh and dry weight by 10.7% and 13.5%, root fresh and dry weight 26.9% and 8.8%, and shoot length by 1.64% under salinity stress, respectively. Moreover, 20 mM Pro treatment more significantly increased growth attributes under salinity stress as well as control condition (Figure 1). For a few growth attributes, cv. Round of pea plant showed more improvements compared to L-888 with foliar application of Pro with and without stress (Figure 1).

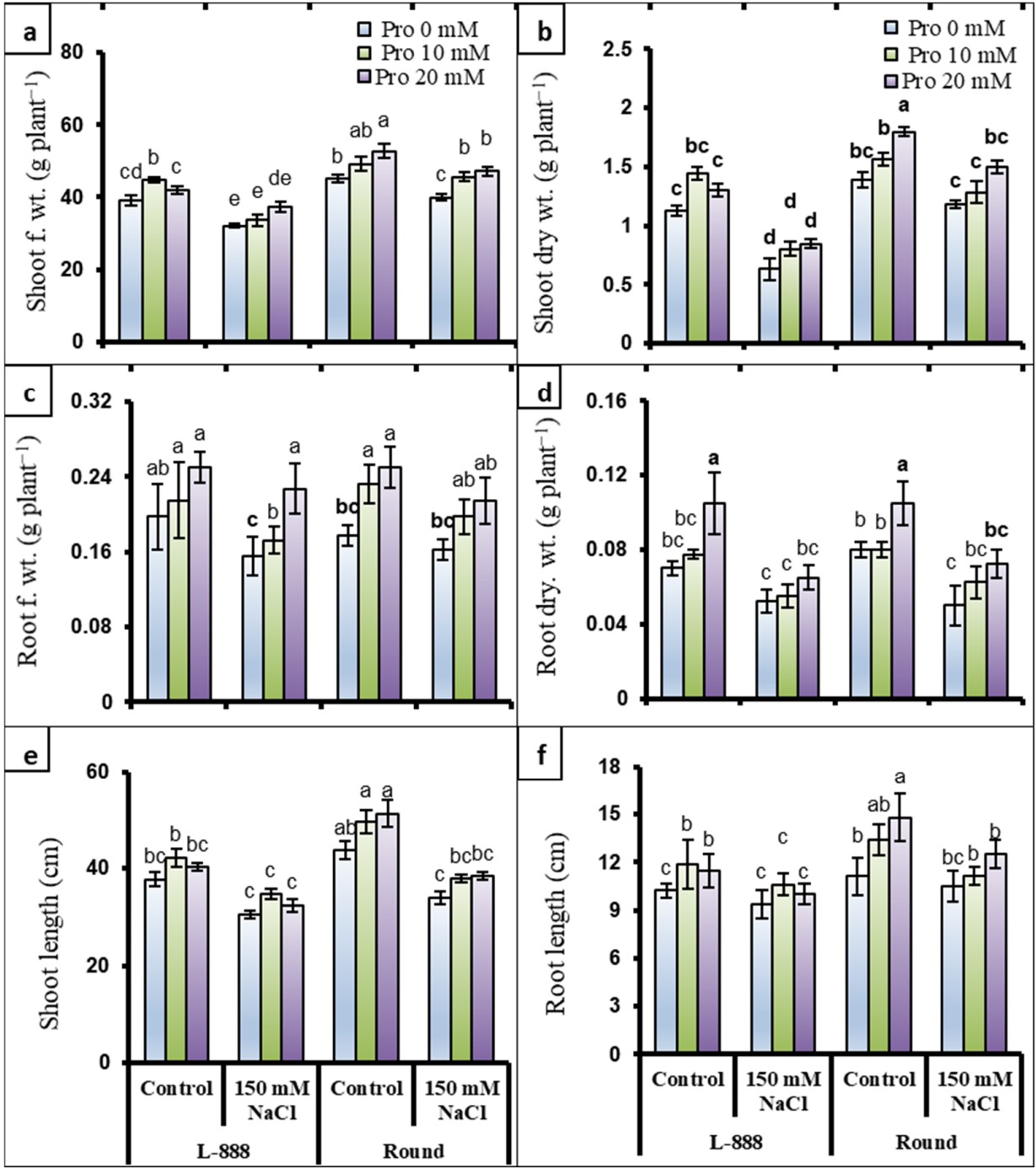

**Figure 1.** Changes in the shoot fresh weight (**a**), shoot dry weight (**b**), root fresh weight (**c**), root dry weight (**d**), shoot length (**e**), and root length (**f**) of the 70 days old seedlings of two pea cultivars grown under different salt concentration conditions with varying levels of proline foliar application. Values are mean $\pm$ SD of three biological replicates. Means sharing a letter for a parameter do not differ significantly at $p \leq 0.05$.

**Table 1.** Mean squares from analysis of variance of the data for various growth attributes, gas exchange, and chlorophyll fluorescence of pea (*Pisum sativum* L.) plants subjected to different concentrations of foliar-applied proline under saline and non-saline conditions.

| Source of Variations | df | Shoot f. wt. | Shoot Dry wt. | Root f. wt. | Root Dry wt. | Shoot Length |
|---|---|---|---|---|---|---|
| Cultivar (Cv) | 1 | 456.3 *** | 1.88 *** | 0.012 * | 0.008 *** | 1072.5 *** |
| Salinity (S) | 1 | 867 *** | 2.167 *** | 0.000ns | 0.000 ns | 453.2 *** |
| Proline (Pro) | 2 | 142.5 *** | 0.327 *** | 0.015 ** | 0.002 *** | 100.4 *** |
| Cv × S | 1 | 23.80 * | 0.218 *** | 0.000 ns | 0.000 ns | 45.8 * |
| Cv × Pro | 2 | 5.137 * | 0.0146 ns | 0.000 ns | 0.000 ns | 3.403 ns |
| S × Pro | 2 | 11.55 ns | 0.075 ** | 0.001 ns | 0.000 ns | 17.21 ns |
| Cv × S × Pro | 2 | 18.77 ns | 0.010 ns | 0.000 ns | 0.000 ns | 1.56 ns |
| Error | 36 | 6.92 | 0.014 | 0.002 | 0.000 | 9.76 |
| **Source of Variation** | **df** | **Root length** | **Number of pods plant$^{-1}$** | **Pod f. wt.** | **Pods weight plant$^{-1}$** | **Number of seeds Pod$^{-1}$** |
| Cultivar (Cv) | 1 | 25.96 * | 945.1 *** | 177.0 *** | 67571.5 *** | 33.33 *** |
| Salinity (S) | 1 | 32.50 ** | 165.0 *** | 1.74 ** | 3452.2 *** | 0.333 ns |
| Proline (Pro) | 2 | 15.50 * | 1.395 ns | 0.491 ns | 270.98 ns | 1.33 ns |
| Cv × S | 1 | 0.88 ns | 0.520 ns | 0.062 ns | 913.01 ** | 3.00 * |
| Cv × Pro | 2 | 1.628 ns | 0.812 ns | 0.274 ns | 199.5 ns | 0.583 ns |
| S × Pro | 2 | 4.718 ns | 0.395 ns | 0.266 ns | 38.18 ns | 0.083 ns |
| Cv × S × Pro | 2 | 0.446 ns | 0.145 ns | 0.065 ns | 11.97 ns | 0.000 ns |
| Error | 36 | 3.98 | 1.548 | 0.181 | 84.49 | 0.583 |
| **Source of Variation** | **df** | **Number of seeds plant$^{-1}$** | **A** | **E** | **A/E (WUE)** | **$g_s$** |
| Cultivar (Cv) | 1 | 75,366.7 *** | 14.97 *** | 0.060 *** | 118.2 *** | 1704.0 *** |
| Salinity (S) | 1 | 10034.0 *** | 16.20 *** | 0.143 *** | 23.03 ns | 4218.7 *** |
| Proline (Pro) | 2 | 298.8 ns | 1.257 ** | 0.004 ns | 8.117 ns | 228.8 ** |
| Cv × S | 1 | 1474.0 ** | 2.655 ** | 0.000 ns | 86.04 ** | 140.0 ns |
| Cv × Pro | 2 | 154.31 ns | 0.716 * | 0.007 ns | 16.66 ns | 2.02 ns |
| S × Pro | 2 | 8.520 ns | 0.004 ns | 0.001 ns | 0.479 ns | 26.7 ns |
| Cv × S × Pro | 2 | 15.27 ns | 0.008 ns | 0.000 ns | 1.740 ns | 14.64 ns |
| Error | 36 | 164.4 | 0.210 | 0.003 | 8.57 | 40.16 |
| **Source of Variation** | **df** | **$C_i$** | **Fv/Fm** | **$q_P$** | **$q_N$** | **NPQ** |
| Cultivar (Cv) | 1 | 14714 *** | 0.023 ns | 0.001 ns | 0.018 * | 0.0216 * |
| Salinity (S) | 1 | 4320 * | 0.000 ns | 0.003 ns | 0.044 *** | 0.165 *** |
| Proline (Pro) | 2 | 2285.9 ns | 0.023 ns | 0.009 ns | 0.0075 ns | 0.015 * |
| Cv × S | 1 | 5104.6 * | 0.03 ns | 0.033 ns | 0.005 ns | 0.187 *** |
| Cv × Pro | 2 | 653.2 ns | 0.001 ns | 0.008 ns | 0.0018 ns | 0.021 * |
| S × Pro | 2 | 404.1 ns | 0.000 ns | 0.000 ns | 0.000 ns | 0.013 * |
| Cv × S × Pro | 2 | 1425.7 ns | 0.004 ns | 0.006 ns | 0.000 ns | 0.000 ns |
| Error | 36 | 870.5 | 0.008 | 0.010 | 0.003 | 0.004 |

df = degrees of freedom; ***, **, and * significant at 0.001, 0.01, and 0.05 levels, respectively; ns = non-significant. $C_i$ = sub-stomatal $CO_2$ conc.; $g_s$ = stomtal conductance; $E$ = transpiration rate; $A$ = net photosynthetic rate; WUE = water use efficiency; *Fv/Fm* = efficiency of photosystem II; $q_P$ = photochemical quenching; $q_N$ = co-efficient of non-photochemical quenching; *NPQ* = non-photochemical quenching.

Yield attributes, number of pods plant$^{-1}$ (Figure 2a), pod fresh weight (Figure 2b), pods fresh weight plant$^{-1}$ (Figure 2c), number of seeds pod$^{-1}$ (Figure 2d), and number of seeds plant$^{-1}$ (Figure 2e) significantly ($p \leq 0.001$) decreased under salt stress (Table 1). The number of pods plant$^{-1}$, pods fresh weight plant$^{-1}$, and number of seeds plant decreased by 2.18-fold (Figure 2a), 3.49-fold (Figure 2b) and 3.96-fold (Figure 2c), and

number seeds pod$^{-1}$ by 12.89% (Figure 2d) and 93.7% (Figure 2e), respectively, compared to the control (Figure 2). Similarly, the effect of proline application was on all studied yield attributes under salt stress; however, it enhanced significantly under non-stress conditions, particularly at 10 Mm concentration (Figure 2). Both cultivars showed significant ($\leq 0.01$) difference for pod weight plant$^{-1}$, number of seeds pod$^{-1}$ and number of seeds plant$^{-1}$, as cv. Round was higher in these measured yield attributes than those of cv. L-888. Foliar application of different proline levels slightly increased the yield attributes (Table 1).

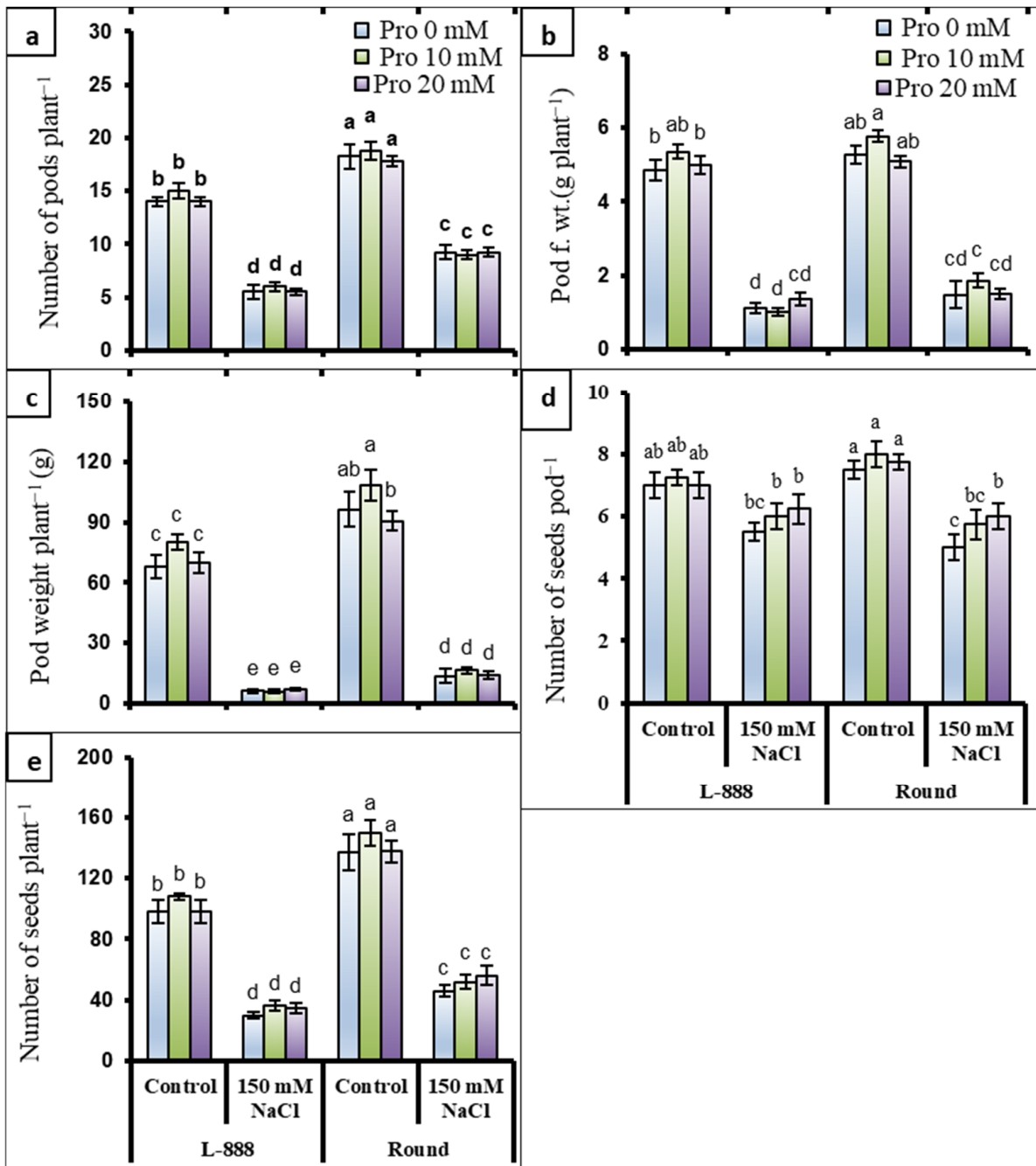

**Figure 2.** Changes in the yield attributes, number of pods plant$^{-1}$ (**a**), pod fresh weight plant$^{-1}$ (**b**), pod weight plant$^{-1}$ (**c**), number of seeds pod$^{-1}$ (**d**), number of seeds plant$^{-1}$ (**e**) harvested at 90 days after sowing of both pea cultivars grown under different salt concentration conditions with varying levels of proline foliar application. Values are mean $\pm$ SD of three biological replicates. Means sharing a letter for a parameter do not differ significantly at $p \leq 0.05$.

### 3.2. Effect of Salinity Stress and Proline on Gas Exchange Characteristics of Pea Seedlings

Net $CO_2$ assimilation rate ($A$), transpiration rate ($E$), water use efficiency (WUE), and stomatal conductance ($g_s$) significantly ($p \leq 0.001$) decreased, while sub-stomatal $CO_2$ ($C_i$) increased in both pea cultivars under salt stress (Table 1). Salinity stress decreased the $A$, $E$, and $g_s$ by 10.29, 9.67, and 8.16%, respectively (Figure 3a,b,d), while $Ci$ increased marginally 2.69% and WUE remain unaltered compared to control (Figure 3c,e). The foliar application of Pro displayed negative impact on WUE by decreasing its efficiency compared to salt stress with Pro treatment (Figure 3c). A considerable variation had been observed in both pea cultivars as cv. Round was higher in gas exchange attributes than those of cv. L-888, except subcellular $CO_2$ concentration ($C_i$) that was high in cv. L-888. Of the varying proline levels, 20 mM showed more stimulating effect on gas exchange characteristics.

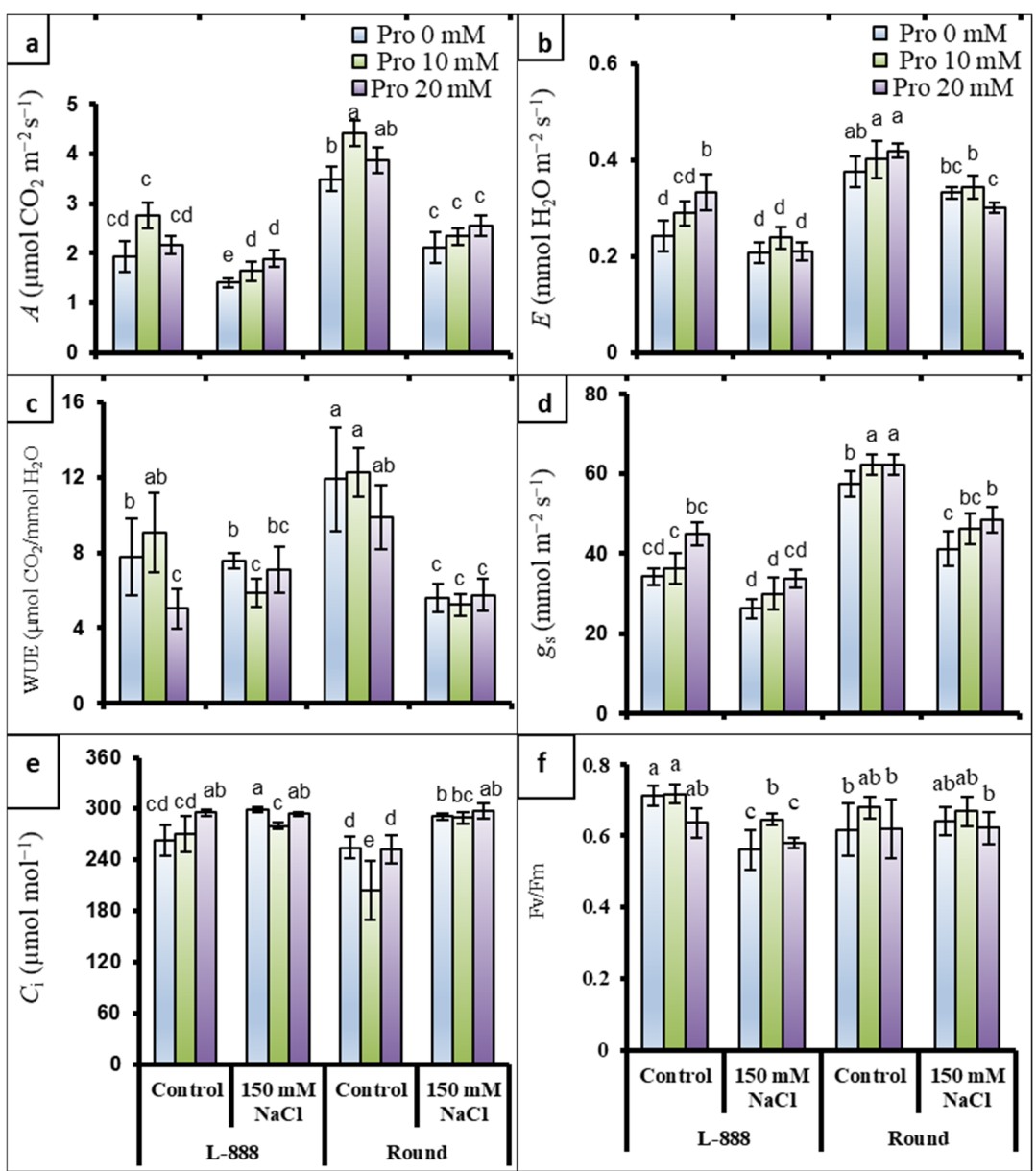

**Figure 3.** Changes of the net photosynthetic rate; $A$ (**a**), transpiration rate; $E$ (**b**), water-use efficiency; *WUE* (**c**), stomatal conductance; $g_s$ (**d**), sub-stomatal $CO_2$ concentration; $Ci$ (**e**), maximum quantum yield (*Fv/Fm*) (**f**) of 70 days old plants of both pea cultivars grown under different salt concentration conditions with varying levels of proline foliar application. Values are mean $\pm$ SD of three biological replicates. Means sharing a letter for a parameter do not differ significantly at $p \leq 0.05$.

### 3.3. Effect of Salinity Stress and Proline on Chlorophyll Fluorescence Attributes of Pea Seedlings

Photosynthetic efficiency might be a determinant factor in identifying the salt stress in both pea cultivars. Maximum quantum yield of photosystem II (*Fv/Fm*) (Figure 3f) and photochemical quenching ($_q$P) (Figure 4a) did not alter either by salt stress or foliar application of various Pro levels (Table 1). However, the coefficient of non-photochemical quenching ($_q$N) (Figure 4c) and non-photochemical quenching (NPQ) (Figure 4d) significantly increased by 28.57% and 31.24% in L-888 pea cultivars under salt stress. Foliar application of proline showed a uniform behavior for $_q$N, while markedly decreased NPQ values in both pea cultivars (Figure 4). Electron transport rate (ETR) (Figure 4b) significantly decreased under both salt stress and foliar application of different proline levels in only pea cultivar Round (Table 2). Of the two proline levels, both levels of proline showed non-significant differences on both pea cultivars (Figure 4).

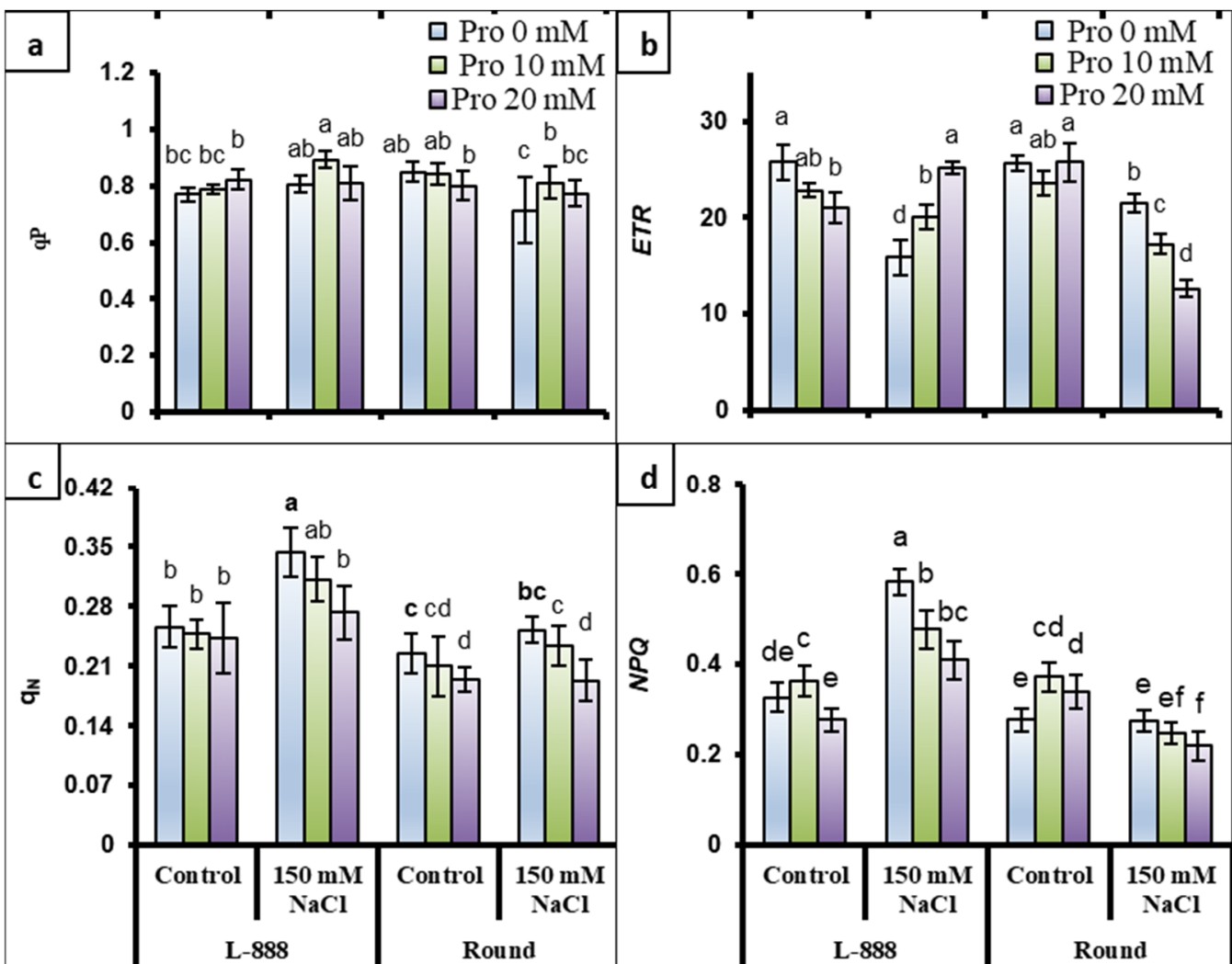

**Figure 4.** Changes of the photochemical quenching; *qP* (**a**), electron transfer rate; ETR (**b**), coefficient of photochemical quenching; *qN* (**c**), non-photochemical quenching; NPQ (**d**) of 70 days-old plants of both pea cultivars grown under different salt concentration conditions with varying levels of proline foliar application. Values are mean ± SD of three biological replicates. Means sharing a letter for a parameter do not differ significantly at $p \leq 0.05$.

**Table 2.** Mean squares from analysis of variance of the data for electron transport rate, total soluble proteins, activities of antioxidant enzymes, free proline, and shoot and root mineral contents of pea (*Pisum sativum* L.) plants subjected to different concentrations of foliar-applied proline under saline and non-saline conditions.

| Source of Variation | df | ETR | Total Soluble Proteins | SOD | CAT | POD |
|---|---|---|---|---|---|---|
| Cultivar (Cv) | 1 | 345.5 *** | 2.354 * | 9.458 ** | 1501.4 *** | 0.561 ns |
| Salinity (S) | 1 | 6.446 ns | 3.840 * | 19.98 * | 113.0 ** | 0.829 ns |
| Proline (Pro) | 2 | 7.18 ns | 1.227 ns | 1.941 ns | 11.66 ns | 2.950 *** |
| Cv × S | 1 | 76.48 ** | 0.964 ns | 1.964 ns | 70.33 * | 3.480 ** |
| Cv × Pro | 2 | 8.138 ns | 1.599 ns | 1.110 ns | 46.75 ns | 1.411 * |
| S × Pro | 2 | 45.27 ** | 0.132 ns | 0.887 ns | 66.33 * | 0.886 ns |
| Cv × S × Pro | 2 | 131.8 *** | 0.235 ns | 0.531 ns | 42.02 ns | 2.850 ** |
| Error | 36 | 6.908 | 0.557 | 1.096 | 14.77 | 0.282 |
| **Source of Variation** | **df** | **Proline** | **Shoot Na$^+$** | **Root Na$^+$** | **Shoot K$^+$** | **Root K$^+$** |
| Cultivar (Cv) | 1 | 31.90 *** | 1104.9 *** | 645.3 *** | 78.79 *** | 338.6 *** |
| Salinity (S) | 1 | 51.41 *** | 373.5 *** | 72.52 ** | 13.54 * | 103.5 *** |
| Proline (Pro) | 2 | 1.377 ns | 5.338 ns | 7.817 ns | 9.41 * | 27.14 *** |
| Cv × S | 1 | 1.397 ns | 540.6 *** | 188.02 *** | 0.421 ns | 10.54 * |
| Cv × Pro | 2 | 1.502 ns | 12.9 * | 6.317 ns | 0.609 ns | 13.64 ** |
| S × Pro | 2 | 0.899 ns | 4.795 ns | 4.098 ns | 0.578 ns | 0.421 ns |
| Cv × S × Pro | 2 | 0.324 ns | 0.124 ns | 17.59 * | 0.484 ns | 0.046 ns |
| Error | 36 | 1.056 | 2.762 | 4.427 | 2.394 | 2.220 |
| **Source of Variation** | **df** | **Shoot Ca$^{2+}$** | **Root Ca$^{2+}$** | | | |
| Cultivar (Cv) | 1 | 73.75 * | 80.08 *** | | | |
| Salinity (S) | 1 | 12.50 * | 20.02 *** | | | |
| Proline (Pro) | 2 | 23.26 * | 10.31 *** | | | |
| Cv × S | 1 | 14.63 * | 0.187 ns | | | |
| Cv × Pro | 2 | 0.817 ns | 0.130 ns | | | |
| S × Pro | 2 | 1.348 ns | 2.442 ns | | | |
| Cv × S × Pro | 2 | 0.505 ns | 0.98 ns | | | |
| Error | 36 | 2.401 | 0.954 | | | |

***, **, and * significant at 0.001, 0.01 and 0.05 levels respectively. ns = non-significant; df = degrees of freedom; ETR = electron transport rate.

### 3.4. Effect of Salinity Stress and Proline on Total Soluble Proteins, Antioxidant Enzymes and Free Proline Contents of Pea Seedlings

Treatment of Pro under NaCl stress resulted in significantly ($p \leq 0.001$) higher leaf concentrations of SOD, CAT and proline (Figure 5) as compared to control plants (Table 2). Total soluble proteins significantly decreased in both pea cultivars under salt stress (Figure 5a). Cultivar Round was higher in soluble proteins as compared to cv. L-888 under saline regimes. Foliar application of proline did not alter free proline contents significantly (Table 2).

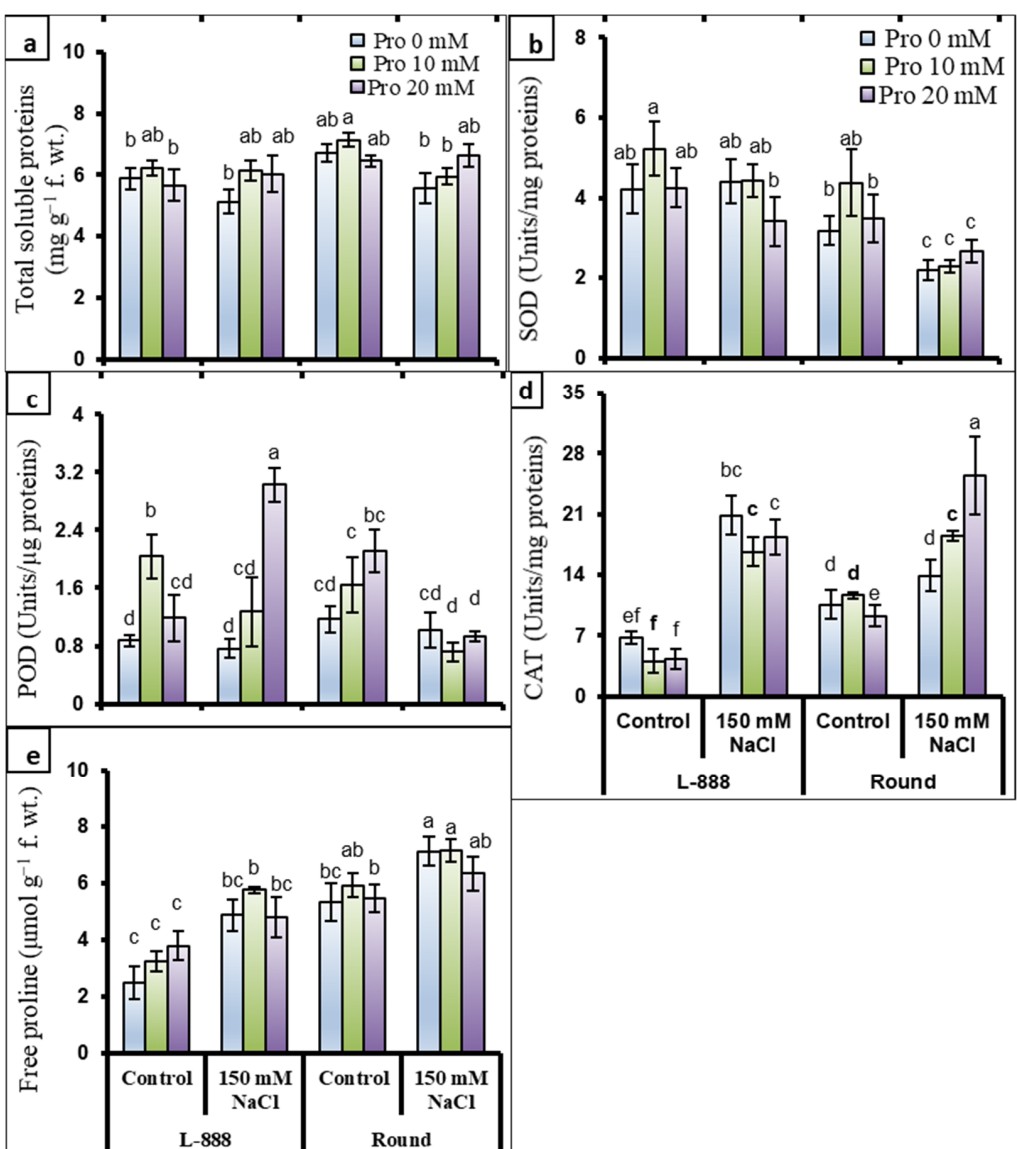

**Figure 5.** Changes of the total soluble protein (**a**), superoxide dismutase (**b**), peroxidase (**c**), catalase (**d**) and free proline (**e**) of 70 days old plants of both pea cultivars grown under different salt concentration conditions with varying levels of proline foliar application. Values are mean ± SD of three biological replicates. Means sharing a letter for a parameter do not differ significantly at $p \leq 0.05$.

Activity of superoxide dismutase (SOD) (Figure 5b) significantly decreased, catalase (CAT) (Figure 5d) increased, while peroxidase (POD) (Figure 5c) remained uniform in both pea cultivars under salt stress. However, exogenous application of 10 mM Pro increased the total soluble protein contents and proline concentration by 10.13%, and 2.3% respectively in cv. L-888. SOD concentration decreased with 20 mM Pro treatment in L-888; however, it increased marginally (2.6%) in cv. Round under salt stress. Of the two pea cultivars, L-888 was higher in SOD activity, while Round excelled L-888 in CAT activity. Foliar application of proline significantly increased POD activity in both pea cultivars, while activities of SOD and CAT were not altered significantly under saline or non-saline conditions (Table 2).

Free proline contents significantly increased under salt stress (Figure 5e). Pea cultivar Round was higher than those of L-888 in accumulation of free proline contents under saline or non-saline conditions (Table 2). Foliar application of proline did not increase the endogenous free proline contents (Table 2).

### 3.5. Effect of Salinity Stress and Proline on Mineral Contents of Pea Seedlings

Imposition of NaCl salinity significantly ($p \leq 0.001$) increased shoot and root Na$^{+1}$ concentration compared to control plants (Table 2). However, application of Pro tended to reduce the Na$^{+1}$ toxicity under salinity treatment (Figure 6). Mineral nutrients (Table 2), including potassium and calcium, were significantly ($p \leq 0.001$) reduced by NaCl salinity treatment compared to control plants. The Shoot and root Na$^{+1}$ concentration increased by 4.08-fold (Figure 6a) and 2.37-fold (Figure 6b), while K$^{+1}$ concentration in shoot and root decreased by 34.16% (Figure 6c,d) and shoot and root Ca$^{2+}$ concentration by 11.19% (Figure 6e,f) compared to the control (Figure 6). However, the foliar application of Pro significantly enhanced shoot and root mineral nutrients with and without salt stress. Cultivar L-888 showed higher shoot and root Na$^{+}$ contents than those of cv. Round, while reverse was true for cv. Round in terms of shoot and root K$^{+}$ and Ca$^{2+}$ contents (Figure 6). Of three levels of proline, 20 mM was proven to be more effective in decreasing Na$^{+1}$, while increasing K$^{+1}$ and Ca$^{2+}$ concentration in both pea cultivars under saline regimes.

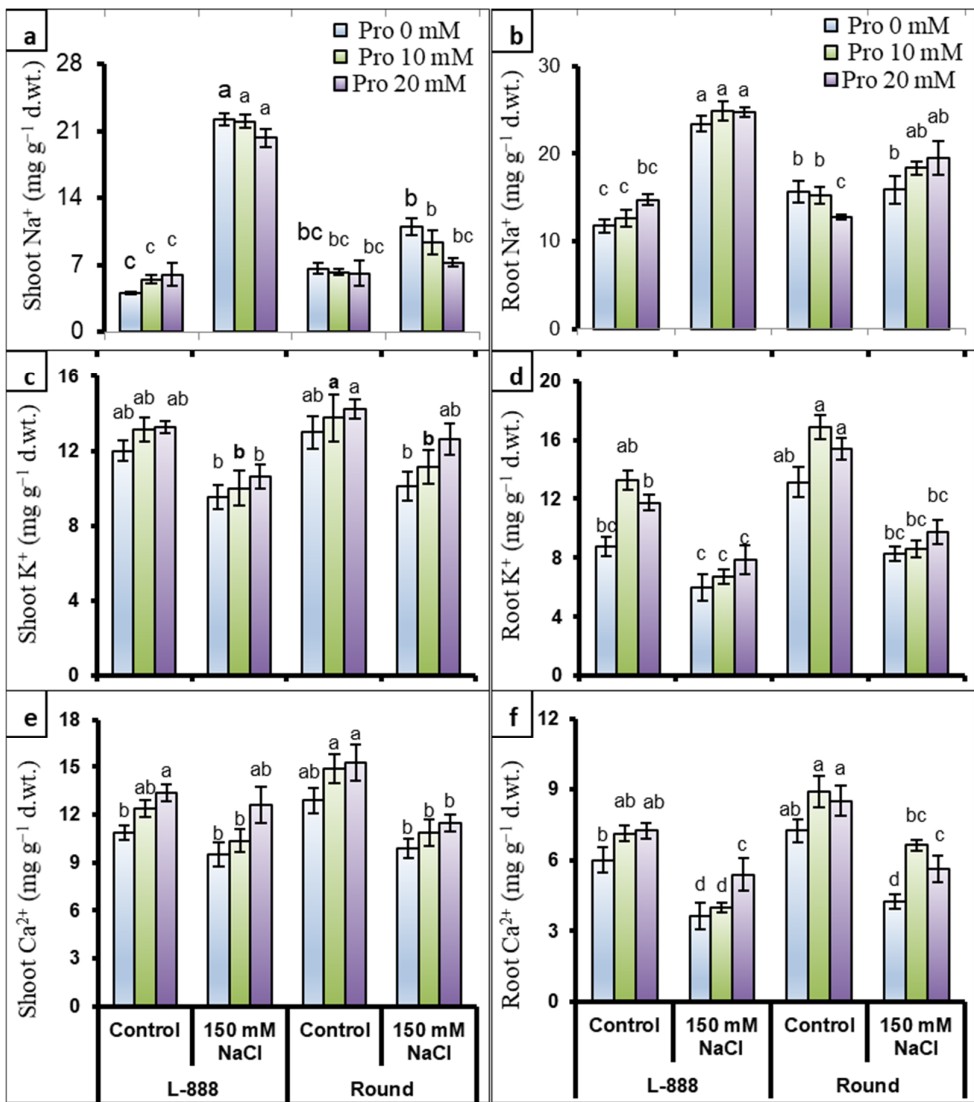

**Figure 6.** Changes of the shoot sodium ions; Na$^{+1}$ (**a**), root sodium ions; Na$^{+1}$ (**b**), shoot potassium ions; K$^{+1}$ (**c**), root sodium ions; K$^{+1}$ (**d**) shoot calcium ions; Ca$^{+2}$ (**e**), and root calcium ions; Ca$^{+2}$ (**f**) of 70-day-oldplants of both pea cultivars grown under different salt concentration conditions with varying levels of proline foliar application. Values are mean ± SD of three biological replicates. Means sharing a letter for a parameter do not differ significantly at $p \leq 0.05$.

## 4. Discussion

Osmotic stress, ionic imbalance and oxidative stress are the major salt-induced mechanisms involved in yield reduction of crop plants [43]. Plants respond to abiotic stresses by the synthesis and accumulation of some low molecular weight metabolites that include various types of free amino acids, particularly proline [44]. Proline comprises 5% of the total pool of free amino acids under normal conditions; however, under stressful conditions its concentration may rise up to 80% in many species [45]. Proline is a compatible osmolyte that plays vital role in osmoregulation by lowering the osmotic potential of cells, allowing more entry of water molecules, thus preventing plant tissues from the desiccation effect caused by water shortage from the external environment. Under stressful conditions proline acts as an osmoprotectant [46];, a reactive oxygen species scavenger [27]; a sink for energy for the regulation of oxidation reduction reactions [47]; protector of macromolecules such as proteins, lipids, nucleic acids from denaturation [48]; a source of nitrogen [49]; and regulates activity of ribulose bisphosphate oxygenase carboxylase (Rubisco). However, some investigators also serve proline as an indicator of salt stress rather than a means of salt stress tolerance [50].

Salt-induced growth and yield reduction in current investigation (Figures 1 and 2), might be an outcome of both reduced stomatal conductance and $CO_2$ fixation along with non-stomatal factors such as impaired photosynthetic machinery [51]. A higher accumulation of salt in the soil causes osmotic imbalance in plants that ultimately effects cell expansion in root tips; thus, it reduces root growth, consequently hindering plant growth and yield [52]. In the current study, foliar application of proline (20 mM) mitigated the salt stress effect on most of the growth and yield attributes of both pea cultivars. In accordance with our findings, exogenous application of proline showed effective role in improving growth under salt stress (Figures 1 and 2) as proven in various crop species such as *Arabidopsis thaliana* [53], tobacco [54], mustard [55], and wheat [56]. Under high-salt conditions, exogenous application of proline significantly enhances plant growth with a marked increase in the seed germination, plant biomass, photosynthesis, gas exchange, and grain yield. Very clearly, these positive stimulatory effects are primarily driven by improved nutrient acquisition, water uptake, and biological nitrogen fixation [57]. Exogenous application of proline may alleviate reduction in growth, shoot lengths, and fresh and dry matter through upregulating stress-protective proteins (i.e., increased synthesis of polypeptides 112 and 48 kDa) under varying levels of salt stress [33].

Low uptake of $CO_2$ ultimately led to a decrease in gas-exchange characteristics, in particular the net $CO_2$ assimilation rate as observed in the current study (Figure 3). However, saline conditions slightly increased sub-stomatal $CO_2$ concentration (Ci). Ionic imbalance and toxic effects of sodium and chloride ions might be another possible reason for reduction in net photosynthetic rate, stomatal conductance, chlorophyll contents and an increase in intracellular $CO_2$ level under salt stress [58,59]. In this study, foliar application of proline significantly increased net photosynthetic rate ($A$) and stomatal conductance ($g_s$) in both pea cultivars (Figure 3). Proline has been considered effective in reducing oxidative stress and enhancing photosynthetic process under salt stress conditions in mustard [15].

Chlorophyll fluorescence attributes might be served as a nondestructive and noninvasive way to undermine the effects of salt stress on the photosynthetic apparatus [60]. We found that the 150 mM NaCl salt stress regime influenced the chlorophyll fluorescence characteristics of both pea cultivars in comparison to the control (Figure 4). Related to the chlorophyll fluorescence, $F_V/F_M$ is the most sensitive and significant attribute. Maximum quantum yield (*Fv/Fm*), photochemical quenching ($q_P$) and electron transport rate (ETR) have been reported to be adversely affected under salt stress [61,62]. In another report, saline stress initially increased the non-photochemical quenching (NPQ), but long-term saline stress decreased it [63]. Hamani [64] suggested that more intensive studies are required to further analyze energy partitioning in response to foliar spraying with osmolytes in salt-stressed plants. Furthermore, some reports also suggested toxic effects of exogenously applied proline when supplied at higher concentrations [26,65]. Similarly in the

current experiment, a higher concentration of Pro tends to decrease some attributes like total soluble sugar, SOD, qP, qN, NPQ, Fv/Fm in pea seedlings under salt stress.

Exogenous application of proline could decrease $H_2O_2$ contents and increase activities of antioxidant enzymes (POD, CAT and ascorbate peroxidase; APX) under salt stress, though it did not improve growth significantly. In this study, SOD activity decreased, while that of CAT (in both pea cultivars) and POD (in cv. L-888 only) increased under salt stress. The cell suspension culture of a tobacco plant also showed significant reduction of the activities of SOD, CAT and POD under salt stress. The foliar application of Pro alleviated the inhibitory effects in CAT and POD activities but not SOD activity under salt stress [66]. Foliar application of proline significantly increased activity of POD in both pea cultivars under both salt stress conditions. In the current study, total soluble proteins decreased under salt stress, while foliar spray of proline did not modulate protein contents significantly under saline or non-saline regimes (Figure 5). Antioxidant enzymes such as SOD protect crop plants by converting superoxide anion to $H_2O_2$, while $H_2O_2$ are decomposed by the action of CAT and POD under salt stress [67].

In the current study, proline contents significantly increased under salt stress in both pea cultivars. Proline accumulation may be linked with decreased protein synthesis, proline utilization, or proteins hydrolysis [68]. It has been reported that proline catabolism is decreased under osmotic stress, with the withdrawal of stress oxidized by proline dehydrogenase that also acts as a pyrroline-5-carboxylate reductase (proline synthesizing enzyme) or proline oxidase (proline degrading enzyme) [44].

Under high-salinity stress toxic sodium and chloride ions interfere in the uptake of water and essential nutrients [69] resulting in osmotic stress and dehydration of cells [70,71]. The accumulation of $Na^+$ in plant cells constrains potassium ion ($K^+$) uptake, which is essential for plant growth [52]. Under high salt-level, uptake of $K^+$ and $Ca^{2+}$ reduced, while that of $Na^+$ increased [72]. Excess $Na^+$ influx disturbs ion homeostasis, which results in abrupt changes in enzyme activities and oxidative stress [73]. An optimum amount of $K^+$ and $Ca^{2+}$ is essential for the functioning and integrity of cellular membranes [74]. Prevention of $Na^+$ influx and enhancing $K^+$ uptake and/or maintaining $K^+$ homeostasis is involved in salt tolerance in plants [75]. Salt tolerant cultivars maintain a high level of $K^+$ and $Ca^{2+}$ ions under salt stress [76,77]. Calcium ($Ca^{2+}$) acts as a signaling molecule and plays important role in maintaining ions homeostasis or osmotic adjustment [9,78]. Osmotic adjustment through accumulation of cheap osmoticum can increase the photosynthetic rate of and afford high-yield production in plants [79]. Among different stressors [80–88], salt stress is becoming an alarming factor for plant growth and development [11]. Proline protects photosynthetic machinery through maintaining ionic homeostasis under salt-induced oxidative stress [89]. In this study, foliar application of proline significantly increased shoot and root $K^+$ and $Ca^{2+}$ contents in both pea cultivars. Cultivar Round showed higher contents of shoot and root $K^+$ and $Ca^{2+}$ as compared to L-888, while the reverse was true for cultivar L-888 in terms of shoot and root $Na^+$ contents.

## 5. Conclusions

In conclusion, salt stress of 150 mM NaCl adversely affected growth, gas exchange characteristics, chlorophyll fluorescence, and mineral nutrients in both pea cultivars. Foliar application of varying proline levels significantly increased growth, yield, net photosynthetic rate ($A$), stomatal conductance ($g_s$), activity of peroxidase, and shoot and root $K^+$ and $Ca^{2+}$ contents, while decreasing non-photochemical quenching values in both pea cultivars. Of the two pea cultivars, cv. Round showed higher growth, yield, photosynthetic efficiency, soluble proteins, activity of catalase, free proline, and shoot and root $K^+$ and $Ca^{2+}$ contents as compared to cv. L-888. Irreversible damage at the cellular level and photoinhibition due to the high production of reactive oxygen species (ROS) and less $CO_2$ availability is directly linked with salinity stress. The detrimental role of exogenously applied proline in mitigating the inhibitory effect of salt stress appears to be dose-dependent. It is yet not clear how proline actually works in minimizing the negative effects of salinity in pea

plants, and further intensive research is needed. Omics approaches can be very helpful in obtaining more holistic molecular perspective of plants compared to other available traditional approaches. To further our understanding of which exogenous proline improves plant–water relations under salt stress, the effect of this osmoprotectant on the expression of aquaporin genes under salt stress will be interesting to investigate.

**Author Contributions:** Conceptualization; S.S. and M.S. Data curation; M.S., M.F.M. and U.Z. Formal analysis; T.J. and M.F.A. Investigation; S.S. Methodology; F.F. and T.J. Project administration; M.S. Resources; M.S. Software; F.F. Supervision; M.S. Validation; M.S. Writing—original draft; S.S. Writing—review & editing; S.S., M.S., F.F., T.J., M.F.A., M.A., A.S.A., M.F.M. and U.Z. All authors have read and agreed to the published version of the manuscript.

**Funding:** The authors thank Taif University Researchers Supporting project number (TURSP 2020/257), Taif University, Saudi Arabia for financially supporting the current research.

**Institutional Review Board Statement:** Not applicable.

**Informed Consent Statement:** Not applicable.

**Data Availability Statement:** Not applicable.

**Acknowledgments:** The authors would like to acknowledge Taif University Researchers Supporting project number (TURSP 2020/257), Taif University, Saudi Arabia.

**Conflicts of Interest:** The authors declare no conflict of interest.

## Abbreviations

| | |
|---|---|
| $A$ | net $CO_2$ assimilation rate |
| $E$ | transpiration rate |
| $g_s$ | stomatal conductance |
| $C_i$ | sub-stomatal $CO_2$ concentration |
| qN | coefficient of non-photochemical quenching |
| NPQ | non-photochemical quenching |
| SOD | superoxide dismutase |
| CAT | catalase |
| POD | peroxidase |
| ROS | reactive oxygen species |
| MDA | malondialdehyde |
| WUE | water use efficiency |
| (Fv/Fm) | maximum quantum yield of photosystem II |
| ETR | electron transport rate |

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
