# Peer review of "Proline-Induced Modifications in Morpho-Physiological, Biochemical and Yield Attributes of Pea (Pisum sativum L.) Cultivars under Salt Stress"

_sustainability, doi:10.3390/su142013579_

Round 1

Reviewer 1 Report

Comments and Suggestions for Authors

General comments: The review paper by authors Shahid et al. is an attempt to present Impact of  Proline-induced modifications in physio-morphological, bio-chemical and yield attributes of pea (Pisum sativum L.) cultivars under salt stress. In my opinion the idea behind the paper is good and meeting the scientific standard in terms of technical language, design and implications of result. The English language of article seems better but some typo errors should be checked briefly during the revision. I suggest minor revision for this article and will be happy to review its revised revision. The following suggestions should be considered to improve the quality of the manuscript before its publication in Sustainability.

1.      Title is not satisfactory enough. Please reframe to maintain the essence of the content.

2.      Please check carefully the spacing between words in the whole text.

3.      Please re-check the abbreviations which have been mentioned in the whole manuscript. Please elaborate them at first mention and used consistently thereafter.

4.      Check and mention the scientific and common names in the whole manuscript. Please add scientific names at first mention and used common names consistently thereafter.

5.      Please be checked the whole manuscript, the format should be improved such as there are some unit presentations, put the relevant unit with each value across the manuscript.

6.      Preparation of proline in the materials and methods is not mentioned.

7.      The authors have provided incorrect information. They mentioned 120 mM NaCl in the abstract and 150 mM in the materials and methods. Please double-check it and be cautious during writing to avoid such gross negligence.

8.      On what basis did the authors choose the concentration of NaCl and proline?

9.         L 153-159, the molarities, pH, and amounts of the solutions used in the analysis of free proline content should be described clearly.

10.  The introduction section is well written and in my opinion it sheds light on the problem in a concise manner, but it should be more elaborated. Authors are suggested to write a brief about the adverse conditions arisen due to salt stress by stating some statistical approximation about the lost irritated areas.

11.  Conclusion part is not satisfactory and can be modified. The prospects need to be rewritten because the profound relationship between salt stress and proline stress is not elucidated, so future research directions should be better pointed out.

Minor changes:

Please find the comments in the attached file and revised it accordingly.

Author Response

Reviewer 1

Sr. No.

Reviewer Comments

Author Response

1

 Title is not satisfactory enough. Please reframe to maintain the essence of the content.

Title has been modified according to suggestion.

2

Please check carefully the spacing between words in the whole text.

Spacing between words has been checked.

3

Please re-check the abbreviations which have been mentioned in the whole manuscript. Please elaborate them at first mention and used consistently thereafter

Thanks, abbreviations in whole manuscript has been checked and it has been confirmed that they have been used in full form at first mention and then abbreviations used.

4

Check and mention the scientific and common names in the whole manuscript. Please add scientific names at first mention and used common names consistently thereafter.

Thanks, needful is done.

5

Please be checked the whole manuscript, the format should be improved such as there are some unit presentations, put the relevant unit with each value across the manuscript.

Thanks, done according to suggestion

6

Preparation of proline in the materials and methods is not mentioned

Proline (Mol. wt. 115.11; Sigma Aldrich), was used for making the different proline concentrations (0, 10 and 20 mM + 0.1% tween-20). A stock solution of proline (30mM) was prepared in deionized water, later diluted to the desired concentration for the experiment

7

The authors have provided incorrect information. They mentioned 120 mM NaCl in the abstract and 150 mM in the materials and methods. Please double-check it and be cautious during writing to avoid such gross negligence.

Thank you for the valuable suggestion. Correction has been done.

8

On what basis did the authors choose the concentration of NaCl and proline?

The concentration of NaCl (Ahmad and Jhon, 2005) and Proline (Huang et al., 2009) was designed according to the available literature

9

L 153-159, the molarities, pH, and amounts of the solutions used in the analysis of free proline content should be described clearly.

Thank you, the molarity, pH and amount of solutions have been incorporated.

10

The introduction section is well written and in my opinion it sheds light on the problem in a concise manner, but it should be more elaborated. Authors are suggested to write a brief about the adverse conditions arisen due to salt stress by stating some statistical approximation about the lost irritated areas

Thank you. We have edited the manuscript as per your suggestion.

11

Conclusion part is not satisfactory and can be modified. The prospects need to be rewritten because the profound relationship between salt stress and proline stress is not elucidated, so future research directions should be better pointed out.

Conclusion part has been revised

Minor changes in the attached File

1

it should be morpho-physiological to maintain the essence of the content

Thanks, Title has been modified according to suggestion.

2

write pea in lower case

Done

3

Line 47: space

Corrected

4

5

Reviewer 2 Report

The manuscript " Proline-induced modifications in physio-morphological, bio-2 chemical and yield attributes of pea (Pisum sativum L.) culti-3 vars under salt stress" have valuable and interesting data. However, there are some questions that need to be addressed for clarification

Major comments:

1. There are logical gaps in statistical analysis. Treatments were not compared through standard statistical method. In all figures author should include above bar letters to compare treatments using appropriate statistical methods for example: Duncan’s Multiple Range test, Tuckey test etc. 

2. The descriptions of results needs to be more scientific and should be written in a more interesting and logical way.

3. The discussion should be more in depth, the studies should be compared in more detail, Please revise it with latest refences.

Minor comments:

Minor suggestions/comments are given in the attached file.

Author Response

Sr. No.

Reviewer 2

Reviewer Comments

Author Response

1

There are logical gaps in statistical analysis. Treatments were not compared through standard statistical method. In all figures author should include above bar letters to compare treatments using appropriate statistical methods for example: Duncan’s Multiple Range test, Tuckey test etc. 

Thank you for your valuable suggestions.

Treatments now compared through standard statistical method. In all figures, Duncan’s Multiple Range test (DMRT) used to show significant difference among the treatments

2

The descriptions of results need to be more scientific and should be written in a more interesting and logical way.

The results have been explained logically by comparing different treatments with %age increase or decrease values

3

The discussion should be more in depth, the studies should be compared in more detail, Please revise it with latest refences.

Minor changes in the attached File

1

Repeated sentence, no need to write again.

Thank you, rephrased

2

Please provide latest data/reference

Thank you, latest data added

3

Similar sentence already mentioned in second paragraph "The over accumulation of salinity...........".So no need to write again, please delete it.

The effect of salt stress and proline now has been explained with particular study on pea plant

4

Apart from proline any other compound available that reduce salinity stress. If yes then mention it if not then replace the sentence with " Exogenous application of proline can reduce salt stress (Kausar et al. 2014, .............2014).

Thank you, added the name of other low molecular weight metabolite ‘glycine betaine’

5

Apart from proline any other compound available that reduce salinity stress. If yes then mention it if not then replace the sentence with " Exogenous application of proline can reduce salt stress (Kausar et al. 2014, .............2014).

Thank you. Given specific reference of salt stress

Round 2

Reviewer 2 Report

Authors have incorporated all the suggestions. The manuscript can be accepted after few minor corrections.

Minor comments:

1. Authors should include list of all abbreviations used in this paper for better understanding, just after keywords or before references. 

2. Line No. 52:  2ntagonistic?? Typo error Please correct it

3. Line No. 850: T. Soleble Proteins??? Please check spelling  

Author Response

  1. Authors should include list of all abbreviations used in this paper for better understanding, just after keywords or before references. 

Response: Esteemed reviewer, abbreviations has been included in manuscript as per your suggestion.

  1. Line No. 52:  2ntagonistic?? Typo error Please correct it

Response: Corrected

  1. Line No. 850: T. Soleble Proteins??? Please check spelling  

Response: Thank you, corrected.